# Evaluation of the EasyScreen™ ESBL/CPO Detection Kit for the Detection of ß-Lactam Resistance Genes

**DOI:** 10.3390/diagnostics12092223

**Published:** 2022-09-14

**Authors:** Camille Gonzalez, Saoussen Oueslati, Laura Biez, Laurent Dortet, Thierry Naas

**Affiliations:** 1Team “Resist” UMR1184 Immunology of Viral, Auto-Immune, Hematological and Bacterial Diseases (IMVA-HB), INSERM, Faculty of Medicine, University Paris-Saclay, LabEx Lermit, 94270 Le Kremlin-Bicêtre, France; 2Bacteriology-Hygiene Unit, Assistance Publique/Hôpitaux de Paris, Bicêtre Hospital, 94270 Le Kremlin-Bicêtre, France; 3Associated French National Reference Center for Antibiotic Resistance, Carbapenemase-Producing Enterobacteriaceae, 94270 Le Kremlin-Bicêtre, France

**Keywords:** multiplex PCR, real-time detection, carbapenemases, ESBLs

## Abstract

Early detection of multidrug resistant bacteria is of paramount importance for implementing appropriate infection control strategies and proper antibacterial therapies. We have evaluated a novel real-time PCR assay using fluorescent probes and 3base^®^ technology, the EasyScreen^TM^ ESBL/CPO Detection Kit (Genetic Signatures, Newtown, Australia), for the detection of 15 β-lactamase genes (*bla*_VIM_, *bla*_NDM_, *bla*_IMP_, *bla*_OXA-48_, *bla*_KPC_, *bla*_OXA-23_, *bla*_OXA-51_, *bla*_SME,_
*bla*_IMI_, *bla*_GES,_
*bla*_TEM,_
*bla*_SHV_, *bla*_CTX-M,_
*bla*_CMY_, *bla_DHA_*) and colistin resistance *mcr-1* gene from 341 bacterial isolates (219 *Enterobacterales*, 66 *P. aeruginosa* and 56 *A. baumannii*) that were grown on Mueller–Hinton (MH) agar plates. One colony was suspended in provided extraction buffer, which lyses and converts the nucleic acids into a 3base^®^-DNA form (cytosines are converted into uracil, and subsequently thymine during PCR). The converted bacterial DNA is then added to the 6 PCR mixes, with primers for three targets plus one internal control. The EasyScreen^TM^ ESBL/CPO Detection Kit was able to detect the 5-major (NDM, VIM, IMP, KPC, OXA-48) and 2-minor (IMI, Sme) carbapenemases and their variants irrespective of the species expressing them with nearly 100% sensitivity and specificity. With cephalosporinases CMY (82% of sensitivity) and DHA (87% of sensitivity) detection of chromosomally encoded variants was less efficient. Similarly, the chromosomally encoded OXA-51 variants were not consistently detected in *A. baumannii*. Despite being capable of efficiently detecting *bla*_CTX-M-_, *bla*_TEM-_, *bla*_SHV-_ and *bla*_GES_-like genes, the EasyScreen™ ESBL/CPO Detection Kit was not able to distinguish between penicillinases and ESBL-variants of TEM and SHV and between GES-ESBLs and GES-carbapenemases. As GES enzymes are still rare, their detection as an ESBL or a carbapenemase remains important. Detection of *mcr-1* was efficient, but none of the other *mcr*-alleles were detected in the 341 bacterial isolates tested. The EasyScreen^TM^ ESBL/CPO Detection Kit is adapted for the detection of the most prevalent carbapenemases encountered in Gram-negatives isolated worldwide.

## 1. Introduction

Antimicrobial agents played a major role in improving the well-being and health of humans all over the world. These molecules are a mainstay of public health [1]. However, while antibiotics have been successful in reducing the burden infectious diseases, their use has increased exponentially, leading to the emergence and spread of antibiotic resistant bacteria. Gram-negative bacteria (GNB), and especially Enterobacterales, *Pseudomonas aeruginosa* and *Acinetobacter baumannii* have emerged as major players in resistance [2]. In these species, resistance may affect all major classes of anti-gram-negative agents (e.g., β-lactams, fluoroquinolones and aminoglycosides), while multidrug resistance (MDR) is relatively common and the rate of infections caused by MDR-GNB is increasing [1,3]. The resistance may in some cases extend to all the available therapeutic agents (leading to a pan-drug resistant phenotype), thus resulting in difficult or impossible to treat infections and turning back the clock to the pre-antibiotic era [1,2]. 

β-Lactams are among the most frequently prescribed antibiotics used to treat bacterial infections due to their safety, reliable killing properties and clinical efficacy. Their utility is, however, threatened by the worldwide proliferation in MDR-GNB of enzymes [β-lactamases (BLs)] capable of inactivating them [4,5]. These BLs are divided into four classes based on their primary amino-acid sequence [4,6]. The emergence of BLs is particularly well illustrated with Enterobacterales resistant to expanded spectrum cephalosporins (ESC) that produce β-lactamases capable of hydrolyzing all β-lactams except carbapenems [7]. The enzymes most frequently involved in ESC resistance are class A extended-spectrum beta-lactamases (ESBL: CTX-M, TEM, SHV) and class C cephalosporinases (either chromosomally or plasmid-encoded: CMY and DHA-2) [8]. As a consequence, an increase in the prescriptions of carbapenems, last resort antibiotics with the broadest spectrum of activity, has been observed [7,8]. Currently, BL-mediated resistance does not spare even the newest and most powerful β-lactams (i.e., carbapenems), whose activity is challenged by carbapenemases belonging either to Ambler classes: A, which include the worldwide disseminated KPC-type enzymes, and to a lesser extend GES, SME, IMI, NMC-A and FRI-1 variants [9,10]; class B or metallo-β-lactamases (MBLs) represented by the three main groups of enzymes such as IMP-, VIM- and NDM-types; and D oxacillinases [carbapenem hydrolyzing class D β-lactamases (CHDL)], with OXA-48 variants found in Enterobacterales, and OXA-23, OXA-24/-40, OXA-58, OXA-143 and the over-expressed intrinsic OXA-51-like enzymes in Acinetobacter spp. and (iii) OXA-198 in *P. aeruginosa* [4,10,11]. The dissemination of carbapenemase-producing (CP) Gram-negatives is a matter of concern, as these pathogens are a major cause of both nosocomial and community-acquired infections, and carbapenems are key players in the treatment of these infections [3]. Colistin has become a ‘last-resort’ drug for treatment of serious clinical infections caused by carbapenem-resistant Enterobacterales [3]. However, increasing use of this antibiotic in clinical and veterinary practice has led to the emergence of plasmid-encoded colistin resistance genes, including mcr-1, which was first reported in *Escherichia coli* in 2015, and mcr-2, identified in *E. coli* in 2016 [12]. The isolation of Enterobacterales expressing both mcr-1 and carbapenemase genes in human clinical samples all over the world raises concerns about the emergence of untreatable bacteria and supports the urgent need for detection methods for both kind of resistance genes [12]. 

MDR-GNB is a major health issues, which requires rapid confirmation of a resistance mechanism not only for implementing effective therapies, but also for infection control measures able to prevent their dissemination [13,14,15]. Most screening protocols are based on cultures of rectal swab specimens on selective media [15], followed by phenotypic confirmation tests to confirm the presence of carbapenem-hydrolyzing activity, such as the Carba NP test and derivatives [16,17], Mace Mast test [18], the carbapenem inactivation method and its derivatives [19], lateral flow immunoassays [20,21,22], imipenem or ESC hydrolysis detected by MALDI-TOF [23], BYG test [24], β-Lacta and β-Carba^TM^ [25,26] or disk diffusion synergy tests (DDST) for detection of MBLs and KPCs (e.g., meropenem disks alone and meropenem disks supplemented with aminophenylboronic acid, dipicolinic acid or cloxacillin) [15]. Culture of rectal swab specimens followed by confirmation testing are long, and often not compatible with rapid implementation of reinforced hygiene measures [17,27]. Molecular-based techniques such as PCR, and whole genome sequencing remain the gold standard for the precise identification of β-lactamase genes [15,28]. Molecular methods are now available for detecting carbapenemase genes from bacterial cultures but also directly from clinical specimens in less than an hour [28]. These tests usually screen either for the five most prevalent carbapenemases [27,29,30,31,32,33] or for some CTX-M-like enzymes [33] but miss minor carbapenemases [10,27,29,30,31,32,33]. 

Here, we have evaluated the performance of the EasyScreen™ Sample Processing and EasyScreen™ ESBL/CPO Detection Kits (Genetic Signatures, Newtown, Australia) for the qualitative detection of nine carbapenemases, four ESBLs and two cephalosporinases widespread in clinically relevant Gram-negatives (NDM, VIM, IMP, KPC, IMI, SME, OXA-48, OXA-23, OXA-51, CMY, DHA, GES, SHV, TEM and CTX-M) and the colistin resistance gene *mcr-1* directly from culture plates.

## 2. Materials and Methods

### 2.1. Media Tested

Bacterial isolates were grown on different media classically used in routine bacteriology laboratories: Trypticase soy agar (TSA), Müller–Hinton (MH) agar, Columbia agar plus 5% horse blood (COH), ChromID^TM^ ESBL agar, ChromID^TM^ CarbaSmart agar plates were from bioMérieux (Marcy-l-Etoile, France) and URISelect^TM^ 4 (Uri4) media were from Bio-Rad (Marne-la-Coquette, France).

### 2.2. EasyScreen™ Sample Processing Kit

The EasyScreen™ Sample Processing Kit is designed to rapidly isolate nucleic acids (DNA and RNA) directly from bacterial culture agar plate or broth. The nucleic acids are converted into 3base^®^ sequences prior to purification/or direct use with the EasyScreen^TM^ ESBL/CPO Detection Kit. The lysis reagent was reconstituted, as recommended by the manufacturer by mixing reagent 1 and 2, and subsequent heating at 80 °C until complete dissolution (Figure 1). Twenty microliters of the combined reagents was dispensed into the provided screw cap tubes (Reaction tube) and one colony, picked using a sterile pipette from culture plates, was resuspended by scraping the tip against the tube and by vortexing. The samples were incubated at 95 °C for 15 min, vortexed and centrifuged before addition of 980 µL of dilution buffer mixed by inversion and centrifugation. For each run, a Negative Process Control (consisting of conversion solution into which a sterile toothpick was dipped) is included. 

### 2.3. EasyScreen™ ESBL/CPO Detection Kit

The converted DNA is subsequently used for PCR amplification using the EasyScreen™ ESBL/CPO Detection Kit, a rapid in vitro nucleic acid amplification assay for the qualitative detection of ESBL and Carbapenemase Producing Organism (CPOs) in nucleic acid from rectal swabs and cultured bacteria. The targets are indicated in Table 1. 

The EasyScreen™ ESBL/CPO Detection Kit includes all reagents required for the detection of the different targets used in the real-time PCR amplification of the nucleic acid, primers and probes labeled with fluorophores as detected by the real-time PCR instrument. In addition, all reaction mixes are manufactured to include an Extraction Control (EC) to determine the reliability of the extracted nucleic acids and indicate the presence of any inhibitors after extraction from primary samples. The 16-plex real-time PCR was performed as recommended by the manufacturer on a Bio-Rad CFX384™ Thermal cycler (Bio-Rad), except for the number of cycles, which was reduced to 35 cycles, using the following protocol: denaturation step of 95 °C for 15 min, 35 cycles (95 °C/2 s; 55 °C/15 s; 60 °C/15 s), and final extension of 65 °C/15 s. The results were interpreted by GS-Call Software provided by the manufacturer. 

### 2.4. Discrepant Results Analysis

Discrepant results between supposed genotype and EasyScreen™ ESBL/CPO PCR results were confirmed using in house PCR, as previously described [20,27,28].

### 2.5. Bacterial Isolates 

Bacterial isolates were from the French National Reference Center (F-NRC) for carbapenem-resistant Enterobacterales strain collection located in the Bacteriology-Hygiene laboratory of the Bicêtre hospital, France. These are representatives of the bacterial isolates circulating in France during the last 10 years and characterized according to the workflow of F-NRC for antimicrobial resistances. A total of 341 clinical isolates comprising 219 Enterobacterales, 66 *Pseudomonas* spp. and 56 *Acinetobacter* spp. harboring single or multiple β-lactamase genes targeted by the assay and/or *mcr-1* genes, and other resistance markers not targeted by the assay for specificity control (Table 2, Table 3 and Table 4). Among the 341 cultured bacteria tested, 189 were carbapenemase-producing GNB and 10 were Mcr-1 producing Enterobacterales. The CP-GNBs consisted in 110 Enterobacterales, among which were 19 KPC-, 6 IMI-, 3 Sme-, 3 GES-, 11 NDM-, 8 VIM-, 8 IMP-, 1 OXA-23, 28 OXA-48-like producers, 15 multiple carbapenemase producers and 5 non-targeted carbapenemase producers (GIM-1, LMB-1, TMB-1, OXA-372 and OXA-58) (Table 2), 45 CP-*P. aeruginosa*, among which were 4 KPC-, 3 GES-like, 16 VIM-like, 8 IMP-like and 7 not-targeted carbapenemases (GIM-1, AIM-1, SPM-1, DIM-1, PME-1 and OXA-198) (Table 3) and 43 CP-*Acinetobacter* spp., among which were 7 OXA-23, 11 NDM, 3 IMP, 6 GES-, 1 VIM, 8 ISAbaI/OXA-51-like and 7 not-targeted carbapenemases (SIM-1, OXA-143, OXA-253, OXA-24/-40, OXA-72, OXA-58, 2 OXA-97) (Table 4). Non-CP-GNB (n = 142) were 109 Enterobacterales, 20 *P. aeruginosa* and 13 *A. baumannii*. 

### 2.6. Statistical Analysis

Performance parameters (sensitivity and specificity) presented here were calculated following resolution of discrepant results, taking the total number of targets as denominator. The sensitivity and specificity values of the EasyScreen^TM^ ESBL/CPO Detection Kit were calculated with their respective confidence interval 95% (95% CI) using the free online software VassarStats: Website for statistical Computation on http://vassarstats.net/ (last accessed on 7 September 2022).

## 3. Results

### 3.1. Experimental Setup and Evaluation of Culture Media

The EasyScreen™ Sample Processing Kits (Reference SP001) lyses any microorganisms present in the sample to be tested and converts the cytosine bases to uracil (detected as thymine after PCR amplification) to create 3base^®^ DNA and RNA. The 3base^®^ nucleic acids lead to increased homology as compared to the native four bases, thus reducing the complexity of genomes. The latter being more similar to each other enables the design of primers and probes with fewer mismatches and that result in better amplification and in less cross-reactivity.

The EasyScreen^TM^ ESBL/CPO Detection Kit has been tested on a large collection (n = 341) of well-characterized clinical isolates expressing various β-lactamases and representing the French epidemiology of CPOs, and of ESC resistant GNB, which is similar to that of many European countries. These isolates belonged to the main multidrug resistant Gram-negatives responsible of infections in human e.g., Enterobacterales (n = 219), *Pseudomonas* spp. (n = 66) and *Acinetobacter* spp. (n = 56) harboring single or multiple β-lactamase genes and/or plasmid-encoded colistin resistant *mcr-1* genes (Table 2, Table 3 and Table 4). The markers detected by this assay are either carbapenemases: *bla*_KPC-like_, *bla*_NDM-like_, *bla*_VIM-like_, *bla*_IMP-like_, *bla*_OXA-48-like_, *bla*_IMI,_
*bla*_Sme_, *bla*_OXA-23_, *Acinetobacter*-related *bla*_OXA-51_, *bla*_GES_; ESBLs: *bla*_GES_, *bla*_CTX-M-like_, *bla*_SHV-like_, *bla*_TEM-like_, penicillinases: *bla*_SHV-like_, *bla*_TEM-like_; cephalosporinases *bla*_CMY-like_, *bla*_DHA-like_ and plasmid-encoded colistin resistant *mcr-1* genes.

Five CPOs (1 OXA-48-producing *E. coli*, 1 KPC-producing *K. pneumoniae*, 1 IMP-producing *K. pneumoniae*; 1 VIM-producing *P. aeruginosa*; and 1 NDM-OXA-23 producing *A. baumannii*) were grown on six different culture media (MH, TSA, Uri4, COH, ChromID ESBL and CARBA Smart) and were tested to see whether they are compatible with the extraction procedure. There was no difference between these media in terms of amplification (data not shown). Thus, during this retrospective evaluation, the assay was used as recommended by the manufacturer, on fresh overnight bacterial colonies grown on MH plates. 

The procedure was easy to perform, taking approximately 16 min for one isolate and up to 45 min for 24 isolates for sample preparation, 2 min for one isolate and up to 60 min for 24 isolates for plate preparation, and a run time of approximately 90 min for 35 cycles using a the CFX 384 instrument (Bio-Rad). The results were interpreted by GS-Call Software (Genetic Signatures) and presented as positive or negative for a given gene with Ct values.

As the PCR were conducted on colonies, the number of cycles was reduced to 35, as preliminary results showed late Cts (>40), which likely corresponded to contamination due to high DNA concentrations. With 35 cycles, this problem was solved as no false or unexplained positive PCRs were recorded. 

### 3.2. Evaluation of the EasyScreen^TM^ ESBL/CPO Detection Kit on Well Characterized Enterobacterales

For the Enterobacterales, a collection of 17 isolates including wildtype isolates or isolates producing non-targeted β-lactamases were tested (TMB-1, GIM-1, FRI-1, OXA-58, ACC-1, ACT-like, over-expressed AMPC). None of these isolates gave a positive PCR result, except one E. coli-producing TEM-1. Two *Citrobacter freundii* isolates producing LMB-1 or TMB-1 remained negative even for CMY, which is the chromosome-encoded AmpC of *C. freundii* (Table 2). 

The EasyScreen^TM^ ESBL/CPO Detection Kit detected all KPC (n = 23), NDM (n = 22), VIM (n = 14), IMP (n = 8), and OXA-48 variants (n = 44). All OXA-48 variants with carbapenemase activity (n = 41), including OXA-162, -181, -204, -232, -244, -370, -484, -505, -517, -519, -535 and -793 were accurately detected. However, OXA-163 and OXA-405, which lack significant carbapenem-hydrolysis (n = 3), were falsely detected as OXA-48-like carbapenemase. Thus, the assay displays a very good specificity and sensitivity for the five main carbapenemases (nearly 100% specificity and sensitivity for every gene sought). Three additional increasingly isolated carbapenemases, Sme-like (n = 3), IMI/NmcA-like (n = 6) and OXA-23 (n = 1), were also detected by the EasyScreen^TM^ ESBL/CPO Detection Kit with a specificity and a sensitivity of 100%, respectively. Finally, GES carbapenemases (n = 3) were also detected, but the assay could not discriminate between GES-ESBLs and GES-carbapenemases. Thus, taken together, carbapenemase detection by the EasyScreen^TM^ ESBL/CPO Detection Kit revealed a sensitivity and a specificity of 100.00% (CI95: 97.00% to 100.00%) and 96.30% (CI95: 90.79% to 98.98%), respectively. These values are higher than those of competitors, as the EasyScreen^TM^ ESBL/CPO Detection Kit also detects emerging carbapenemases such as IMI, SME, OXA-23 and GES enzymes [30].

A large panel of CTX-M variants (n = 100) were accurately detected, such as CTX-M-1, CTX-M-9, CTX-M-15, CTX-M-24, CTX-M-55, CTX-M-71, except 2 CTX-M-8-producers, which remained negative, despite repeated attempts. Thus, the EasyScreen^TM^ ESBL/CPO Detection Kit detects four out of the five main CTX-M groups but fails to detect CTX-M-8 variants, which are increasingly isolated in the southern Americas [34,35]. Molecular diagnostic tests targeting CTX-M have already been described, but only CTX-M-1 or CTX-M-9 groups were detected [33].

The EasyScreen^TM^ ESBL/CPO Detection Kit accurately detected nearly all SHV (n = 65/68) and TEM (n = 107) variants. It also detected all ESBL variants (SHV-5, -12, -27, -99, and TEM-24, -3 and -52), but was not able to distinguish between penicillinase and ESBL variants of TEM and SHV. Indeed, SHV was systematically detected in all K. pneumoniae strains, even in those lacking an SHV-ESBL. Thus, considering ESBL detection, the EasyScreen^TM^ ESBL/CPO Detection Kit revealed a sensitivity and a specificity of 95.76% (CI95: 90.39% to 98.61%) and 40.98% (CI95: 35.01% to 47.15%), respectively.

The assay was able to detect all plasmid-encoded DHA and CMY-variants tested, but failed to identify some chromosome-encoded variants, especially with CMY-enzymes. Indeed, chromosome-encoded variants of CMY-150, CMY-135 and CMY-13 were not detected, while plasmid encoded CMY-136, -16, -6, -48, -4 and -2 were reproducibly detected. Only 3/14 *C. freundii* isolates tested positive, while 13/13 plasmid-encoded blaCMY-genes were detected. Thus, considering CMY detection, the EasyScreen^TM^ ESBL/CPO Detection Kit revealed a sensitivity and a specificity of 68.57% (CI95: 50.71% to 83.15%) and 100% (CI95: 98.02% to 100.00%), respectively.

Finally, the EasyScreen™ ESBL/CPO Detection Kit is also able to detect mcr-1 colistin resistance genes with a specificity of 100%. However, it was not able to detect other MCR variants such as MCR-2, -3, -4 and -5.

### 3.3. Evaluation of the EasyScreen^TM^ ESBL/CPO Detection Kit on Characterized Pseudomonas *spp.* Isolates

A collection of 22 Pseudomonas spp. negative controls gave negative PCR results (MEX C/D- OprJ, OprD deficient, CARB-4, BEL, VEB-1, PER-1, GIM-1, AIM-1, SPM-1, PME-1, DIM-1) (Table 3). 

The EasyScreen^TM^ ESBL/CPO Detection Kit was able to detect all KPC-, NDM- and VIM-producers. With IMP-producers, while most IMP variants, including IMP-2, -15, -31, -63, -71 were accurately detected, two variants, IMP-13 and IMP-29, were reproducibly not detected. Similarly, the assay detected all GES-producers (n = 4) but was not able to distinguish between GES-carbapenemase and GES-ESBLs. Thus, considering carbapenemase detection in *P. aeruginosa* isolates, the EasyScreen^TM^ ESBL/CPO Detection Kit revealed a sensitivity and a specificity of 92.11% (CI95: 78.62% to 98.34%) and 96.30% (CI95: 81.03% to 99.91%), respectively.

As for Enterobacterales, all CTX-M, SHV and TEM-producers were detected. For SHV, both isolates tested expressed an SHV ESBL, while for TEM only one TEM-ESBL was present among the five tested positive TEM-producers. Thus, considering ESBL detection in *P. aeruginosa* isolates, the EasyScreen^TM^ ESBL/CPO Detection Kit revealed a sensitivity and a specificity of 100.00% (CI95: 63.06% to 100.00%) and 92.98% (CI95: 83.00% to 98.05%), respectively. 

The other resistant determinants were not present in the panel of tested isolates.

### 3.4. Evaluation of the EasyScreen^TM^ ESBL/CPO Detection Kit on Characterized A. baumannii Isolates

For the *Acinetobacter* spp. evaluation, a collection of 15 negative control isolates that do not express any of the acquired resistance genes have been tested (WT, PER-1, VEB-1, SCO-1, RTG-4, SIM-1, OXA-21, OXA-40, OXA-58, OXA-253 and OXA-143). None of these non-isolates gave positive PCR results, except for 6/15 OXA-51 and 3 TEM-positive PCR results (Table 4). 

Among *A. baumannii*, all IMP-, VIM-, NDM- and OXA-23 -producers were detected (Table 4). All the GES variants were detected irrespective of their hydrolysis profile. Considering that only 8/13 GES-variants tested were carbapenemases, an overall sensitivity and a specificity of 100.00% (CI95: 85.18% to 100.00%) and 84.85% (CI95: 68.10% to 94.89%), respectively, were determined for carbapenemase detection in *P. aeruginosa* isolates using the EasyScreen^TM^ ESBL/CPO Detection Kit. However, for chromosome-encoded OXA-51, only 37 out of the 55 OXA-51-like alleles were identified. 

Similarly, 23 *bla*_TEM_ genes were detected, while none corresponded to a TEM-ESBL. Only two SHV-5 ESBLs and two CTX-M-producers were detected. 

### 3.5. Overall Assay Performance for Carbapenemase Detection Extrapolated to the French Epidemiology of CP-GNB

When extrapolating these results to the global French epidemiology, as represented by the carbapenemase genes identified by the French National Reference Center (F-NRC) for Carbapenem-resistant Enterobacterales, *P. aeruginosa* and *A. baumannii*, excellent specificity and sensitivity might be expected for the detection of CPEs (Table 5) [36]. Among the 9468 CPE isolates sent to the F-RC for CPEs between 2012 and 2018, the performances of the EasyScreen^TM^ ESBL/CPO Detection Kit would have been 99.97% sensitivity (95%CI: 99.91–99.99) and 99.99% specificity (95%CI: 99.95–100.00) with one OXA-405-producer being falsely identified as CPE and three non-targeted CPEs would have been missed-1 FRI-1 and 2 TMB-1-producers [30]. Similarly, when extrapolating our results to the global French epidemiology of CP *P. aeruginosa*, the EasyScreen^TM^ ESBL/CPO Detection Kit performances would have been 98.22% sensitivity (95%CI: 94.9–99.6) and 100.00% specificity (95%CI: 99.3–100.00), missing only 3 DIM-producers. For *Acinetobacter* spp., when extrapolating our results to the global French epidemiology of CP *A. baumannii*, the EasyScreen^TM^ ESBL/CPO Detection Kit performances would have been 88.25% sensitivity (95%CI: 84.5–91.4) and 100.00% specificity (95%CI: 75.3–100.00), missing only 38 OXA-24/40 and 5 OXA-58-producers.

Taken together, our results obtained with EasyScreen^TM^ ESBL/CPO Detection Kit on our collection of isolates, and when extrapolated to the global French epidemiology of CPOs, we would expect (in the best way) 99.97% sensitivity for CPE detection, 98.22% for CP Pseudomonas spp. and 88.25% for CP-Acinetobacter spp. These results are better than those of BD MAX Check-Points CPO Assay [30], which only detects the five main carbapenemases, with sensitivities of 99.26% for CPE detection, 93% for CP *Pseudomonas* spp. and only 12.5% for CP-*Acinetobacter* spp.

## 4. Discussion

The EasyScreen™ ESBL/CPO Detection Kit is an efficient and rapid (<3 h from colony to result) detection system for the most frequently encountered penicillinases (including ESBLs), carbapenemases, cephalosporinases and MCR-1 in Enterobacterales, *Pseudomonas* spp. and *A. baumannii*. Concerning the evaluation of *Pseudomonas* spp. and *Acinetobacter* spp. strains, we observed more false-positive and negative results and consequently the percentage of specificity and sensitivity is lower especially with CMY, IMP, GES and OXA-51 variants. 

The EasyScreen™ ESBL/CPO Detection Kit was performed on cultured bacteria using a boiling extraction protocol, which requires hands on time. However, the EasyScreen™ ESBL/CPO Detection Kit is compatible with most existing automated nucleic acid extraction and real-time PCR instruments and can thus be fully automated.

Irrespective of the host bacteria, the EasyScreen™ ESBL/CPO Detection Kit showed excellent biological performances (sensitivity and specificity) for the five most common carbapenemases, including IMP variants that constitute a very heterogeneous family of enzymes and that are not well detected by most molecular assays [27]. The results of our study revealed that the EasyScreen™ ESBL/CPO Detection Kit is well adapted to the French epidemiology of CPE and CP-*Pa,* which reflects that of many countries in Europe and around the world. The short turnaround time and its simplicity make it suitable for routine use in clinical microbiology laboratories. It can provide results from colonies that grew on most commonly used culture plates, including MH and complex chromogenic and selective screening media. 

The kit also detects the main carbapenemases encountered in Enterobacteriaceae (KPC, GES, NDM, VIM, IMP, OXA-48-like) in *P. aeruginosa* (VIM, IMP, NDM and KPC) and in *A. baumannii* (OXA-23, VIM, IMP, NDM), except minor OXA-carbapenemases usually identified in *A. baumannii* (OXA-24/-40-like and OXA-58-like) that are not targeted by this assay. When extrapolated to the global French epidemiology of CPOs, we would expect (in the best way) 99.97% sensitivity for CPE detection, 98.22% for CP *Pseudomonas* sp. and 88.25% for CP-*Acinetobacter* spp. Thus, the Younden index for the French situation would be 99.27, 93.3, 88% for CP-E, CP-*Pa* and CP-*Ab,* respectively. 

Minor carbapenemases are increasingly isolated and often responsible for outbreaks, as no commercially available assay targets these resistance determinants [37,38,39,40,41,42,43,44]. At the French national reference center in 2020, 22 strains of *Enterobacter cloacae* complex producing class A carbapenemases IMI/NMC-A, 11 strains of *Proteus* spp. producing OXA-23 and one strain of *K. pneumoniae* producing GES-5 have been identified, representing 1.1% of all carbapenemases received during that year [37]. Recently, in a multicentric study performed in 12 French hospitals, 26.9% (14/52) of the amoxicillin-clavulanate-resistant *Proteus mirabilis* isolates produced an OXA-23 carbapenemase, suggesting that this resistant determinant might be underestimated in France [38]. As early as 2013, Bush et al. have shown the silent spread of Sme enzymes in the United States and suggested these carbapenemase to be included in carbapenemase detection systems. Recently, Public Health England has revealed increasing identification of *S. marcescens*-producing Sme carbapenemases in the United Kingdom [39,40]. Similarly, GES-5 carbapenemase-producing GNB have increasingly been described in multispecies outbreaks on different continents [41,42,43,44]. In most of these cases, the index patient was missed as no commercially available assay targeted this carbapenemase, and epidemiological follow up was difficult, as it relied on home-made PCR, or on WGS which is costly and time consuming [41]. 

Similarly, it is the only commercially available assay that allows detection of ESC resistance by targeting the main ESBL, CTX-M-like, but also some minor ESBLs (VEB prevalent in South-East Asia, PER in Turkey, and GES in Greece and in the Arabic Peninsula), which turns this assay into a globally adapted assay [45]. Finally, it includes the two main plasmid-encoded AmpCs, currently spreading in most countries [36]. Regarding TEM and SHV, as it is not able to distinguish between penicillinase and ESBL variants, these results should be considered only when all the other ESBLs genes are negative, and yet the bacteria display a classical ESBL phenotype. In a recent study conducted in France, out of 100 consecutively isolated ESBLs, 98 were CTX-M variants and 2 *E. cloacae* were SHV-2a [20]. These latter using the EasyScreen™ ESBL/CPO Detection Kit were positive for SHV and displayed an ESBL phenotype on disc diffusion antibiogram. Its simplicity and short turnaround time (less than 2 h for one sample) makes it suitable for use in the routine microbiology laboratory. It can provide results from colonies that grew on MH but also from selective screening media.

The 3base^®^ nucleic acids reduces the complexity of genomes allowing the design of primers and probes with fewer mismatches, which results in better amplification and in less cross-reactivity.

Mutation and/or polymorphisms in the primer/probe binding region of the targeted gene may lead to false negative PCR results and thus to non-detection of some variants. The use of the 3base^®^ nucleic acids is less prone to mutations/polymorphisms, as it reduces the complexity of genomes allowing the design of primers and probes with fewer mismatches, which results in better amplification and increased detection of variants. This is best exemplified with IMP variants, a very heterogeneous family of carbapenemases. Nevertheless, as for every molecular assay, they must be evaluated on isolates corresponding to the local epidemiology in-order-to know which are the alleles that might be missed, especially in a constantly evolving field of carbapenemases. The penultimate goal is now the evaluation of the EasyScreen™ ESBL/CPO Detection Kit directly on clinical samples, especially on rectal swabs to identify carriers of carbapenemase/ESC resistance genes, and the use of Genetic Signatures’ full automation solutions would be ‘good to have’ in order to speed up extraction and plate preparation. 

## Figures and Tables

**Figure 1 diagnostics-12-02223-f001:**
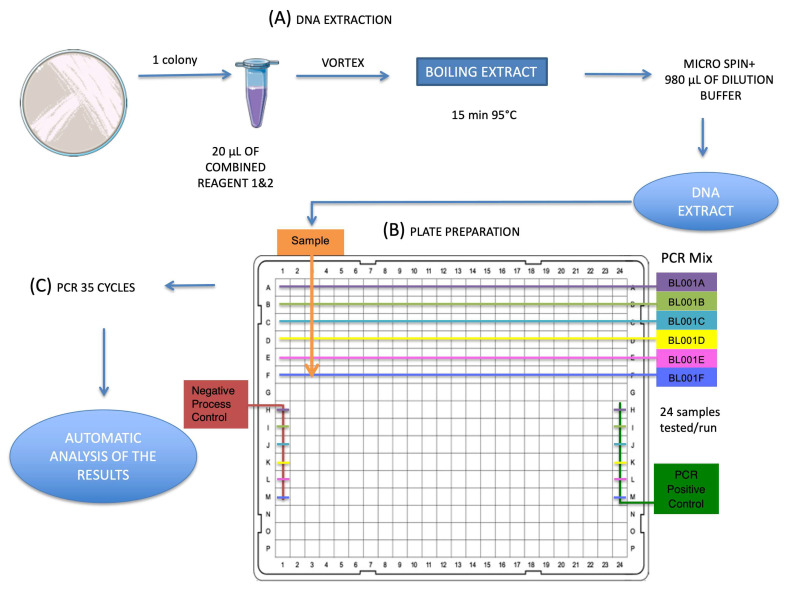
Experimental procedures. (**A**) Complete sample processing (lysis of any microorganisms present in one colony and conversion of all cytosine bases into uracil (detected as thymine after PCR amplification) with combined Reagent 1 and 2. (**B**) Two microliters of DNA extract together with sixteen microliters of PCR mixes was added to each well of the 384-well microplate. (**C**) Real-time PCR assay (EasyScreen™) on CFX384™. PCR amplification and result interpretation using the manufacturer’s proprietary software (GS-Call Software). Total duration of the assay for one sample: c.a. 1h50 up to 3h15 for 24 samples.

**Table 1 diagnostics-12-02223-t001:** PCR Layout.

Detection Channel	Mix A	Mix B	Mix C	Mix D	Mix E	Mix F
# 1	TEM	GES	OXA-48	MCR-1	VIM	CMY
# 2	EC ^a^	EC ^a^	EC ^a^	EC ^a^	EC ^a^	EC ^a^
# 3	CTX-M	KPC	OXA-23-like	DHA	IMI	IMP
# 4	Not used	NDM	SHV	OXA-51-like	SME	Not used

^a^: Extraction Control.

**Table 2 diagnostics-12-02223-t002:** Global performances of the EasyScreen^TM^ ESBL/CPO Detection Kit on Enterobacterales isolates colonies grown on MH agar (n = 219).

Species	Resistance Mechanism		EasyScreen^TM^ ESBL/CPO Detection Kit Results														
		KPC	OXA-48	NDM	VIM	IMP	SME	IMI	GES	CTX-M	TEM	SHV	DHA	CMY	OXA-23	OXA-51	Mcr-1
**Non-targeted β-lactamase producers (17)** ^a^																	
*E. coli* (2) ^a^*, P. mirabilis (2), E. cloacae* (3) *H. alvei (2), S. marcescens (1)*	↗↗↗ ^b^ Case (7) ^a^, pACC-1 (3)	- ^c^	-	-	-	-	-	-	-	-	1/1	-	- ^c^	-	-	-	-
*E. cloacae* (3), *P. mirabilis* (1), *C. freundii* (2)	GIM-1, LMB-1, TMB-1, OXA-372, OXA-58	-	-	-	-	-	-	-	-	-	1/1	-	-	0/2	-	-	-
**Targeted AmpC producers**																	
*E. coli (3)*, *K. pneumoniae* (2), *M. morganii (1)*	DHA (5), CMY-136	-	-	-	-	-	-	-	-	-	1/1	3/3	5/5	1/1	-	-	-
**Targeted ESBLproducers**																	
*E. coli (32)*, *K. pneumoniae (13), C. freundii (2), C. koseri (1), M. morganii (1), E. cloacae (4), P. mirabilis (1)*	CTX-M-1(5), -2(3), -3(2), -8(2), -9(1), -10(1), -13(1), -14(3), -15(16), -17(1), -18(1), -19(1), -24(1), -27(2), -32(2), -37(1), -55(2), -57(1), -65(1), -71(2), -82(1), -93(1), 101(1), 182(1)		-	-	-	-	-	-		52/54	29/29	11/13	1/1	1/3	-	-	-
*K. pneumoniae*	GES-1,		-	-	-	-	-	-	1/1	-	-	1/1	-	-	-	-	-
*K. pneumoniae*, *E. cloacae, S. marcescens*	OXA-163 (2), OXA-405	-	3/3	-	-	-	-	-	-	-	2/2	2/2	-	-	-	-	-
*K. pneumoniae* (6), *C. freundii*	SHV-2a (2), -11, -12, -36, -38, -99	-	-	-	-	-	-	-	-	1/1	3/3	7/7	-	1/1	-	-	-
*E. coli (1)*, *E. cloacae* (1), *K. aerogenes (2), C. freundii* (2), *K. pneumoniae (1), P. mirabilis*	TEM-3 (4), -24(2), -52 (2)	-	-	-	-	-	-	-	-	-	8/8	1/1	-	0/2	-	-	-
**Targeted carbapenemase producers**																	
*P. mirabilis*	OXA-23	-	-	-	-	-		-	-	-	-	-		-	1/1	-	-
*S. marcescens* (3)	Sme-1, Sme-2, Sme-4	-	-		-	-	3/3	-	-	-	-	-		-	-	-	-
*E. cloacae complex* (6)	IMI-1, IMI-2 (2), IMI-3, IMI-17, NMCA	-	-	-	-	-		6/6	-	-	-	-		-	-	-	-
*K. pneumoniae* (1), *E. cloacae* (2),)	GES-5 (2), GES-6	-	-	-	-	-		-	3/3	1/1	-	2/2		-	-	-	-
*E. coli* (11), *K. pneumoniae* (4), *E. cloacae* (2), *C. freundii, S. marcescens*	KPC-2 (7), KPC-3 (4), KPC-5, KPC-6, KPC-7, KPC-14, KPC-28, KPC-31, KPC-33, KPC-39	19/19	-	-	-	-		-	-	4/4	9/9	4/4		0/1	-	-	-
*E. coli* (9), *K. pneumoniae* (1)*, S. enterica*	NDM-1 (5), NDM-4, NDM-6, NDM-6, NDM-9, NDM-19 (2)	-	-	11/11	-	-	-	-		8/8	9/9	1/1	2/2	4/4		-	-
*E. coli* (2), *K. pneumoniae* (3), *E. cloacae*, *C. freundii* (2)	VIM-1 (4), VIM-2 (2), VIM-4, VIM-19	-	-	-	8/8	-	-	-	-	1/1	3/3	4/4	-	1/3	-		-
*E. coli* (3), *K. pneumoniae* (3), *E. cloacae, S. marcescens*	IMP-1 (2), IMP-8 (4), IMP-10, IMP-IMP-14	-	-	-	-	8/8	-	-	-	-	6/6	6/6	-	-	-	-	
*E. coli* (12), *K. pneumoniae* (11), *E. cloacae, C. koseri, C. freundii (2), Shewanella* sp.	OXA-48 (8), OXA-162, OXA-181 (2), OXA-204 (5), OXA-232 (2), OXA-244 (4), OXA-370, OXA-484, OXA-517, OXA-519, OXA-535, OXA-793		28/28							16/16	15/15	14/14		8/8			
**Targeted multiple carbapenemase producers**																	
*E. coli* (3), *K. pneumoniae* (4), *E. cloacae*, *C. freundii* (2)	OXA-48-like + NDM-Like (6); OXA-48-like + VIM-like (3); OXA-48-like + KPC-28	1/1	10/10	6/6	3/3					8/8	7/7	5/5		0/2			1/1
*K. pneumoniae*	OXA-48-like + NDM-5 + VIM-1		1/1	1/1	1/1							1/1					
*K. pneumoniae (2)*	NDM + KPC (2)	2/2		2/2						1/1	2/2	1/2					
*E. coli*	NDM-1 + VIM-2			1/1	1/1						1/1						
*K. pneumoniae*	KPC-2 +VIM-1	1/1			1/1						1/1	1/1					
**Plasmid-encoded colistin resistance**																	
*E. coli (6), K. pneumoniae, Salmonella (4)*	mgrB mutation (6), mcr-3.2, mcr-4, mcr-5(2), mcr-2.									1/1	3/3	1/1					-
*E. coli (6), K. pneumoniae (3), Salmonella* sp.	Mcr-1	-	2/2	1/1	-	-	-	-	-	5/5	6/6	3/3	-	-	-	-	10/10
Total		23/23	44/44 ^d^	22/22	14/14	8/8	3/3	6/6	4/4 ^e^	98/100 ^f^	107/107 ^g^	65/68 ^h^	8/8	16/27 ^i^	1/1	0/0	10/10
Sensitivity		100% (85.2% to 100%)	100% (91.4% to 100%)	100% (84.6% to 100%)	100% (76.8% to 100%)	100% (63.1% to 100%)	100% (29.2% to 100%)	100% (54.1% to 100%)	100% (29.2% to 100%)	98.0% (92.9% to 99.8%)	7.5% (3.3% to 14.2%	10.3% (4.24% to 20.1%)	100% (63.1% to 100%)	59.3% (38.8% to 77.6%)	100% (2.5% to 100%)		100% (69.2% to 100%)
Specificity		100% (98.1% to 100%	98.3% (95.2% to 99.6%	100% (98.1% to 100%	100% (98.2% to 100%	100% (98.3% to 100%	100% (98.3% to 100%	100% (98.3% to 100%	99.5% (97.5% to 100%	100% (96.9% to 100%	100% (96.7% to 100%	100% (97.6% to 100%	100% (98.3% to 100%	100% (98.1% to 100%	100% (98.3% to 100%	100% (98.3% to 100%	100% (98.3% to 100%

^a^ Number of isolates or resistance mechanism tested; ^b^ ↗↗↗ Case, Abbreviation for overexpressed cephalosporinase; ^c^ no amplification; ^d^ 41 OXA-carbas, 3 non carbas detected as carba; ^e^ 3 GES-carbas, 1 non GES-ESBL detected as carbapenamase; ^f^ 1/2 CTX-M-8 and 1/1 CTX-M-17 were missed by this assay; ^g^ 107 TEM detected, but only 8 were TEM-ESBLs; ^h^ 65/68 SHV producers detected, but only 7 were SHV-ESBLs; ^i^ Only 16/27 CMY producers detected.

**Table 3 diagnostics-12-02223-t003:** Global performances of the EasyScreen^TM^ ESBL/CPO Detection Kit on *Pseudomonas* spp. (n = 65).

Species	β-Lactamase Content		EasyScreen^TM^ ESBL/CPO Detection Kit Results														
		KPC	OXA-48	NDM	VIM	IMP	SME	IMI	GES	CTX-M	TEM	SHV	DHA	CMY	OXA-23	OXA-51	Mcr-1
**Non-targeted β-lactamase producers (22) ^a^**																	
*P. aeruginosa* (2) ^a^	Mex A/B-OprM, Mex C/D-OprJ	- ^c^	-	-	-	-	-	-	-	-	-	-	- ^c^	-		-	
*P. aeruginosa* (4)	↗↗↗ ^b^ Case + OprD deficient	-	-	-	-	-	-	-	-	-	-	-	-	-	-	-	-
*P. aeruginosa* (9)	BEL-1, VEB-1, PER-1, OXA-10, OXA-13, OXA-14, OXA-18/20, OXA-32, CARB-2	-	-	-	-	-	-	-	-	-	-	-	-	-	-	-	-
*P. aeruginosa* (7)	GIM-1, AIM-1, SPM-1, DIM-1 (2), PME-1, OXA-198	-	-	-	-	-	-	-	-	-	-	-	-	-	-	-	-
**Targeted ESBLproducers**																	
*P. aeruginosa*	GES-9,		-	-	-	-	-	-	1/1	-	-	-	-	-	-	-	-
*P. aeruginosa*	CTX-M-2	+	-	-	-	-	-	-	-	1/1	-	-	-	-	-	-	-
*P. aeruginosa* (2)	SHV-2a, SHV-5	+	-	-	-	-	-	-	-	-	-	2/2	-	-	-	-	-
*P. aeruginosa*	TEM-4	+	-	-	-	-	-	-	-	-	1/1	-	-	-	-	-	-
**Targeted carbapenemase producers**																	
*P. aeruginosa* (3)	GES-2, GES-5, GES-14	-	-	-	-	-		-	3/3	-	-	-		-	-	-	-
*P. aeruginosa* (4)	KPC-2 (4)	4/4	-	-	-	-		-	-	-	-	-		-	-	-	-
*P. aeruginosa* (2)	NDM-1 (2)	-	-	2/2	-	-	-	-			-	-	-	-		-	-
*P. aeruginosa* (7), *P. putida 2), P. stutzeri, P. fluorescens*	VIM-1 (2), VIM-2 (7), VIM-4 (1), VIM-5 (1)	-	-	-	11/11	-	-	-	-	-	2/2	-	-	-	-		-
*P. aeruginosa* (12), *P. putida (2), P. stutzeri*	IMP-1 (3), IMP-2(2), IMP-7, IMP-15, IMP-19, IMP-26, IMP-31, IMP-39, IMP-46, IMP-56, IMP-63, IMP-71	-	-	-	-	15/15	-	-	-	-	2/2		-	-	-	-	
*P. aeruginosa* (3),	IMP-13 (2), IMP-29	-	-	-	-	0/3	-	-	-		-	-	-	-	-	-	-
**Total**		**4/4**	**-**	**2/2**	**11/11**	**15/18 ^d^**			**4/4 ^e^**	**1/1**	**5/5 ^g^**	**2/2 ^f^**	**-**	**-**	**-**	**-**	**-**
Sensitivity		100% (39.8% to 100%)		100% (15.8% to 100%)	100% (71.5% to 100%)	83.3% (58.6% to 96.4%)			100% (29.2% to 100%)	100% (2.5% to 100%)	100% (2.5% to 100%)	100% (15.8% to 100%)					
Specificity		100% (94.1% to 100%)	100% (94.5% to 100%)	100% (94.3% to 100%)	100% (93.4% to 100%)	100% (92.4% to 100%)	100% (94.5% to 100%)	100% (94.5% to 100%)	98.4% (91.3% to 99.9%)	100% (94.4% to 100%)	93.8% (84.8% to 98.3%)	100% (94.3% to 100%)	100% (94.5% to 100%)	100% (94.5% to 100%)	100% (94.5% to 100%)	100% (94.5% to 100%)	100% (94.5% to 100%)

^a^ Number of isolates or resistance mechanism tested; ^b^ ↗↗↗ Case, Abbreviation for overexpressed cephalosporinase; ^c^, no amplification.; ^d^ IMP-13 and IMP-29 producers not detected; ^e^ ¾ GES variants were carbapenemases; ^f^ 1/5 was a TEM ESBL; ^g^ 2/2 SHV ESBL detected.

**Table 4 diagnostics-12-02223-t004:** Global performances of the EasyScreen^TM^ ESBL/CPO Detection Kit on *Acinetobacter* spp. (n = 56).

Species	β-Lactamase Content		EasyScreen^TM^ ESBL/CPO Detection Kit Results														
		KPC	OXA-48	NDM	VIM	IMP	SME	IMI	GES	CTX-M	TEM	SHV	DHA	CMY	OXA-23	OXA-51	Mcr-1
**Non-targeted β-lactamase producers (15) ^a^**																	
*A. baumannii* (8) ^a^	WT (2), OXA-21, PER-1, VEB-1 (2), SCO-1, RTG-4,	- ^b^	-	-	-	-	-	-	-	-	2/2	-	- ^c^	-	-	2/8	-
*A. baumannii* (7)	SIM-1, OXA-143, OXA-253, OXA-24/40, OXA-72, OXA-58, OXA-97	-	-	-	-	-	-	-	-	-	1/1	-	-	-	-	4/7	-
**Targeted ESBLproducers (5)**																	
*A. baumannii*	GES-12,	-	-	-	-	-	-	-	1/1	-		-	-	-	-	1/1	-
*A. baumannii* (2)	CTX-M-2, CTX-M-15	-	-	-	-	-	-	-	-	2/2	2/2	-	-	-	-	2/2	-
*A. baumannii*	SHV-5	-	-	-	-	-	-	-	-	-	1/1	1/1	-	-	-	1/1	-
*A. baumannii*	TEM-1	-	-	-	-	-	-	-	-	-	1/1	-	-	-	-	1/1	-
**Targeted carbapenemase producers (36)**																	
*A. baumannii* (6)	GES-11 (3), GES-14 (3)	-	-	-	-	-		-	7/7	-	1/1	-		-	1/1	7/7	-
*A. baumannii* (8)	ISAba1/OXA-51-like (3), ISAba1/OXA-51-like + SHV-5, ISAba1/OXA-51-like + GES-12 (4)	-	-	-	-	-		-	4/4	-	3/3	1/1		-	-	2/8	-
*A. baumannii* (7)	OXA-23 (5), OXA-23 + GES-1, OXA-23 +GES-14	-	-	-	-	-		-	1/1	-	4/4			-	6/6	6/6	-
*A. baumannii* (11)	NDM-1 (4), NDM-1 + OXA-23, NDM-1 (4) + ISAba1/OXA-51-like, NDM-2, NDM-9	-	-	11/11	-	-		-	-	-	2/2	-		-	6/6	11/11	-
*A. baumannii (3)*	IMP-1 (2), IMP-4 + OXA-58	-	-	-	-	3/3	-	-	-	-	3/3	-	-	-	-	0/3	
*A. genomospecies 16*	VIM-4	-	-	-	1/1	-	-	-	-	-	-	-	-	-	-	0/1	-
**Total**		**-**	**-**	**11/11**	**1/1**	**3/3**	**-**	**-**	**13/13 ^c^**	**2/2**	**23/23 ^d^**	**2/2 ^e^**	**-**	**-**	**13/13**	**37/55 ^f^**	
Sensitivity				100% (71.5% to 100%)	100% (2.5% to 100%)	100% (29.2% to 100%)			100% (63.1% to 100%)	100% (15.8% to 100%)					100% (75.3% to 100%)	67.3% (53.3% to 79.3%)	
Specificity		100% (93.6% to 100%)	100% (93.6% to 100%)	100% (92.1% to 100%)	100% (93.5% to 100%)	100% (93.3% to 100%)	100% (93.6% to 100%)	100% (93.6% to 100%)	89.6% (77.3% to 96.5%)	100% (93.4% to 100%)	64.6% (51.7% to 76.1%)	96.9% 8(9.3% to 99.6%)	100% (93.6% to 100%)	100% (93.6% to 100%)	100% (91.8% to 100%)		100% (93.6% to 100%)

^a^ Number of isolates or resistance mechanism tested; ^b^ no amplification; ^c^ 8/13 GES variants were carbapenemases; ^d^ none was a TEM ESBL; ^e^ 2/2 SHV ESBL detected; ^f^ 37/55 OXA-51 identified.

**Table 5 diagnostics-12-02223-t005:** Global performances of the EasyScreen^TM^ ESBL/CPO Detection Kit extrapolated to the global French epidemiology of CP-GNBs.

	EasyScreen^TM^ ESBL/CPO Detection Kit		
	Global French CP*E* Epidemiology ^a^	Global French CP-*Pa* Epidemiology ^b^	Global French CP-*Ab* Epidemiology ^c^
	(2012–2018)	2018	2018
**Isolates received at the F-NRCs**	**19,600**	**678**	**379**
**Carbapenemase positive isolates**	**9468** (6798 OXA-48, 284 KPC, 1676 NDM, 492 VIM, 123 IMP, 54 IMI; 1 FRI-1; 2 GES; 10 OXA-23; 2 TMB-1; and 1 SME-4)	**169** (127 VIM, 1 NDM, 0 KPC, 18 IMP, 20 GES, 3 DIM)	**366** (284 OXA-23, 38 OXA-24/40 et 5 OXA-58; 65 NDM)
**Sensitivity**	**99.97% (95%CI: 99.91–99.99)**	**98.22% (95%CI: 94.9–99.6)**	**88.25% (95%CI: 84.5–91.4)**
**Specificity**	**99.99% (95%CI: 99.95–100)**	**100% (95%CI: 99.3–100)**	**100% (95%CI: 75.3–100)**
**Expected False positive**	1 OXA-405	None	None
**Expected False negative**	1 FRI, 2 TMB-1 => 3 isolates	3 DIM, =>3 isolates	38 OXA-24/40 and 5 OXA-58-producers => 43 isolates

Carbapenemase-producing ^a^ Enterobacterales, ^b^
*P. aeruginosa* and ^c^ A. *baumannii.*

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
