# Peer review of "Evaluation of the EasyScreen™ ESBL/CPO Detection Kit for the Detection of ß-Lactam Resistance Genes"

_diagnostics, 2022, doi:10.3390/diagnostics12092223_

Round 1
Reviewer 1 Report
The manuscript evaluated the performance of the EasyScreen ESBL/CPO PCR Kit for the detection of 15 ESBL, AmpC, carbapenemase, and mcr-1 genes from 341 gram-negative isolates.
1. The "Methods" described that five different media (MH, TSA, Uri4, ChromID ESBL and Carba Smart) were used and the "Results" indicated that six different media (MH, TSA, Uri4, Blood, ChromID ESBL and Carba Smart) were used. Which one is correct?
2. Are the results compared by sub-culturing all the 341 isolates in several different media (5 or 6) , respectively? Or is it just a partial comparison? The authors stated that there was no difference in the results, but it was 100% consistent? If possible, it would be better to describe specific numbers. For example, 99% or more.
Author Response
- The "Methods" described that five different media (MH, TSA, Uri4, ChromID ESBL and Carba Smart) were used and the "Results" indicated that six different media (MH, TSA, Uri4, Blood, ChromID ESBL and Carba Smart) were used. Which one is correct?
Answer : Thank you for pointing out this discrepancy. Indeed six media were tested. This has been corrected in the marterial and methods section (line 111-115).
- Are the results compared by sub-culturing all the 341 isolates in several different media (5 or 6) , respectively? Or is it just a partial comparison? The authors stated that there was no difference in the results, but it was 100% consistent? If possible, it would be better to describe specific numbers. For example, 99% or more.
Answer : As the results with the different media were comparable (within the same Ct), the testing with all the strains was done using colonies grown on MH media. This has been clearly indicated in the text (222-228).
Reviewer 2 Report
The authors developed a novel real-time PCR assay based on the EasyScreen™ ESBL/CPO Detection Kit for the detection of multiple resistant genes. Result showed that this assay is efficient for detecting the five most common carbapenemases with nearly 100% sensitivity and specificity. The study might be potentially interesting, I suggest the paper should be accepted after addressing the following concerns.
1. I suggest author should apply their proposed assay in the real sample
2. The sensing properties such as sensitivity and selectivity for detecting clinically-relevant Gram-negatives and the colistin resistance gene should be compared to the previously reported ones.
3. Please keep the format of reference consistent.
Author Response
The authors developed a novel real-time PCR assay based on the EasyScreen™ ESBL/CPO Detection Kit for the detection of multiple resistant genes. Result showed that this assay is efficient for detecting the five most common carbapenemases with nearly 100% sensitivity and specificity. The study might be potentially interesting, I suggest the paper should be accepted after addressing the following concerns.
Answer: thank you for your encouraging words
- I suggest author should apply their proposed assay in the real sample
Answer: We agree that this is the ultimate goal. However, the assay is only validated on pure cultures. Testing real clinical samples is more complicated, as we have to apply for ethical approval, which takes at least 6 months, in France. This will be the matter of a future work, once all the legal aspects have been complied to.
- The sensing properties such as sensitivity and selectivity for detecting clinically-relevant Gram-negatives and the colistin resistance gene should be compared to the previously reported ones.
Answer: This can done for one single target, but not for the entire test, as this test detects many more targets, and especially targets that are not detected by any other commercially available assay. Discussion on specificity is added for the main targets (Line: 378-381 for CPE; line 395-396 for minor CPEs; line 412-414)
- Please keep the format of reference consistent.
Answer: References have been checked throughout